
# AMV Error Characterization and Bias Correction by Leveraging Independent Lidar Data: a Simulation using OSSE and Optical Flow AMVs

Hai Nguyen[1], Derek Posselt[1], Igor Yanovsky[1], Longtao Wu[1], and Svetla Hristova-Veleva[1]

[1]Jet Propulsion Laboratory, California, 4800 Oak Grove Dr, Pasadena, CA

**Correspondence:** Hai Nguyen (hai.nguyen@jpl.nasa.gov)

**Abstract.**

Accurate estimation of global winds is crucial for various scientific and practical applications, such as global chemical transport modeling and numerical weather prediction. One valuable source of wind measurements is Atmospheric Motion Vectors (AMVs), which play a vital role in the global observing system and numerical weather prediction models. However, errors in AMV retrievals need to be addressed before their assimilation into data assimilation systems, as they can affect the accuracy of outputs.

An assessment of the bias and uncertainty in passive-sensor AMVs can be done by comparing them with information from independent sources such as active-sensor winds. In this paper, we examine the benefit and performance of a colocation scheme using independent and sparse lidar wind observations as a dependent variable in a supervised machine learning model. We demonstrate the feasibility and performance of this approach in an Observing System Simulation Experiment (OSSE) framework, with reference geophysical state data obtained from high resolution Weather Research and Forecasting (WRF) Model simulations of three different weather events.

Lidar wind data are typically available in only one direction, and our study demonstrates that this single component of wind in high-precision active-sensor data can be leveraged (via a machine learning algorithm to model the conditional mean) to reduce the bias in the passive-sensor winds. Further, this active-sensor wind information can be leveraged through an algorithm that models the conditional quantiles to produce stable estimates of the prediction intervals, which are helpful in design and application of error analysis. We also found that the uncertainty prediction of this single-component wind has a positive linear relationship with the total-vector root-mean-squared-vector-difference (RMSVD), which can aid in design of quality indicators and filters.





## 1 Introduction

The accurate estimation of global winds is critical for various scientific and practical applications, including global chemical transport modeling and numerical weather prediction. One source of wind measurements is atmospheric motion vectors (AMVs), which are obtained through the tracking of cloud or water vapor features in satellite imagery. They play a crucial role in the global observing system, providing essential data for initializing numerical weather prediction (NWP) models; these AMVs are particularly valuable for constraining the wind field in remote Southern Hemisphere regions and over the world's oceans, where other wind observations are scarce. Obtaining global measurements of three-dimensional winds was emphasized as an urgent need in the NASA Weather Research Community Workshop Report (Zeng et al., 2016) and identified as a priority in the 2007 National Academy of Sciences Earth Science and Applications from Space (ESAS 2007) decadal survey, as well as in ESAS 2017. Numerous studies have demonstrated the positive impact of AMVs on the forecast accuracy of global NWP models (Bormann and Thépaut, 2004; Velden and Bedka, 2009; Gelaro et al., 2010). Further uses include studying global CO2 transport (Kawa et al., 2004), providing inputs for weather and climate reanalysis studies (Swail and Cox, 2000), and estimating present and future wind-power outputs (Staffell and Pfenninger, 2016). Major NWP centers now incorporate AMVs from various geostationary and polar-orbiting satellites, resulting in nearly global horizontal coverage, though vertical resolution is generally quite coarse.

Numerical weather prediction integrates Atmospheric Motion Vectors (AMVs) through a process called data assimilation, which involves combining observations of atmospheric variables with an a priori estimate of the atmospheric state (usually generated by a short-term forecast) to derive a posterior estimate of wind fields and other atmospheric state variables. To achieve accurate results, each input source of information is weighted using an inverse error covariance matrix meant to represent the accuracy of the data. Nguyen et al. (2019) analytically proved that inaccurate error characterizations of the inputs (i.e., a priori information) can adversely affect the bias and validity of the outputs, and similarly it is important to assess, and if possible, correct for biases in AMVs retrievals before their subsequent usage in data assimilation. Staffell and Pfenninger (2016), for instance, observed that NASA's MERRA and MERRA-2 wind product suffer significant spatial bias, overestimating wind output by 50% in northwest Europe and underestimating by 30% in the Mediterranean, and they noted that such biases can have adverse effect on the quality of data assimilation that ingests said data. Therefore, it is of paramount importance to assess and remove the biases inherent in AMV retrievals before their usage in subsequent analysis.

In practice, correcting the bias of an AMV retrieval requires an independent proxy for the 'truth', and previous studies assessing AMV uncertainty typically compared AMVs with collocated radiosonde data and AMVs derived from Observing System Simulation Experiments (Cordoba et al., 2017). Here, we propose the idea of using the independent (and sparse) lidar observations of wind as a dependent variable in a supervised machine learning model for bias correction. Following the OSSE framework of Posselt et al. (2019), we examine a proof-of-concept that demonstrates the feasibility and performance of an bias-correction scheme in an Observing System Simulation Experiment (OSSE) framework. We use as our reference (truth, or NatureRun) datasets output from the Weather Research and Forecasting (WRF) Model run for three different weather events (Posselt et al., 2019). The water vapor fields from these WRF model runs are processed through an Optical Flow algorithm



(Yanovsky et al., 2023) to provide AMVs, and we assess the ability of a bias-correction algorithm to model and correct biases that arise from the optical flow AMV retrieval.

Velden and Bedka (2009) along with Salonen et al. (2015) have highlighted the significant impact of height assignment on the uncertainty of Atmospheric Motion Vectors (AMVs) derived from cloud movement and sequences of infrared satellite radiance images. However, this error source is intertwined with uncertainties in the water vapor profile itself, and modeling

this within the OSSE framework requires extensive knowledge and parameterization of the height-assignment error process, which is beyond the scope of this paper. As such, in this paper we will focus on fixed-height errors in the AMV estimates and the bias-corrections arising therefrom.

One challenge with pairing passive-sensor and active-sensor winds is that the latter typically observes only in one direction, along the instrument's line of sight. Therefore, a question one might ask is what sort of information a researcher might be able

to obtain on the entire wind-vector if, for example, lidar winds are only available at sparse locations in only the line-of-sight direction. In this paper, we search for the answer to this question in an OSSE framework, and we show that passive sensor can benefit from coincident active sensor data through algorithms that model the expectation (bias reduction) or quantiles (uncertainty quantification). Furthermore, we shall show that having the lidar line-of-sight wind shall allow us to make reasonable estimates of the full wind vector error (i.e., RMSVD).

We are not aware of a similar approach in the literature for leveraging lidar wind retrievals for improvement of AMV retrievals. Perhaps the closest would be Teixeira et al. (2021), which combined random forest with Gaussian mixture models to form regime-based estimates of bias and uncertainty. While this approach in principle can be used to bias-correct observations, it discretizes the bias error function into a fixed number of clusters. While this discretization is useful for understanding the geophysical regimes of the underlying atmospheric processes, it is not as efficient as a model that is purposely built for bias-

minimization.

The intention of this paper is not to propose that the algorithms outlined here should replace error characterization methods for all AMVs. Instead, our primary objective is to demonstrate that residual error patterns exist in AMV retrieval algorithms, regardless of whether they involve traditional feature tracking or optical flow. Furthermore, through meticulous variable selection and algorithm refinement, it is feasible to curtail these biases. We also provide evidence that, in the four selected scenarios,

the confidence intervals predicted using the Meinshausen and Ridgeway (2006) approach exhibit predominantly positive linear correspondence with the empirical validation standard error. This correlation is a notable and valuable characteristic that carries implications for devising indicators of AMV quality.

For the remainder of this paper, we will discuss the data sources, study regions, and the optical flow AMV retrieval algorithm in Section 2. In Section 3.1, we discuss the process of variable selection, and we discuss parameter optimization and bias-

reduction performance in Section 3.2. We follow this treatment of bias with a discussion of modelling uncertainty via prediction intervals in Section 3.3. Finally, we end with some discussion of the merits of our approach and plans for further studies.



| | Region | Spatial Resolution | Temporal Resolution | Beginning Time | Duration |
|---|---|---|---|---|---|
| ETC | Western Atlantic Ocean | 4 km | 120 sec | 2006-11-22 00:02:00 | 12 hr |
| TC | Southeast Asia | 3.5 km | 72 sec | 2008-07-10 06:00:00 | 4.5 hr |
| Harvey EDS | Gulf of Mexico | 3 km | 120 sec | 2017-08-23 18:00:00 | 12 hr |
| Harvey LDS | Gulf of Mexico | 3 km | 120 sec | 2017-08-24 06:00:00 | 12 hr |

**Table 1.** Overview of spatial and temporal parameters for the study scenarios

## 2 Data sources

The evaluation of the impact of bias-correction on optical flow AMV will be carried out in the context of an OSSE. All OSSEs share these key components: 1) A reference dataset, used as a basis for comparison. In our case, this is a NatureRun (NR), which is a high-fidelity simulation mimicking real-world conditions; 2) Simulators generating synthetic observations as if they were taken from the NR (this includes radiative transfer models, retrieval system simulations, and accounting for measurement errors, spatial, and temporal aspects); and 3) A quantitative methodology to evaluate information in the candidate measurements (Posselt et al., 2022). In this section, we shall discuss our choices for these components, with emphasis on the choice of study regions, the water vapor retrieval simulations, and the algorithm for computing AMV from the water vapor.

### 2.1 Study regions

For a comprehensive view of the impact of bias-correction across various atmospheric scenarios, we will examine three different systems which include an extratropical cyclone, tropical convection, and a hurricane at its early and late development stages. The details of these datasets describing the storm systems are summarized in Table 1. The extratropical cyclone (ETC) reference scenario is one that developed east of the United States in the western Atlantic Ocean in late November 2006. This cyclone showcases a diverse spectrum of wind speeds, water vapor contents, and gradients, providing an extensive assessment of the AMV algorithm's error traits across a wide array of atmospheric circumstances. This specific ETC case is selected due to its thorough examination in numerous previous observational studies (Posselt et al., 2008; Crespo and Posselt, 2016). For this case, simulations are carried out using the Advanced Research Weather Research and Forecasting (WRF) Model, version 3.8.1 (Skamarock et al. 2008). The model is configured with three nested domains (d01, d02, and d03) operating at horizontal resolutions of 20, 4, and 1.33 km, respectively, although for this paper we will focus primarily on the data at 4 km resolution and at pressure levels at 850, 500, and 300 hPa. For our analysis, we focus primarily on the 12-hour span of the storm starting on 2006-11-22 00:02:00 UTC.

For the tropical convection case, we consider a simulated water vapor dataset over the Maritime Continent from 0600 UTC to 1027 UTC on 10 July 2008. Similar to the ETC scenario, we chose pressure levels at 850, 500, and 300 hPa for analysis. Further details on this simulation can be found in Yanovsky et al. (2023).





Our third study scenario is a hurricane NatureRun, produced by initializing the Weather Research and Forecasting (WRF) Model using initial conditions from a ensemble forecast of Hurricane Harvey and described in further details in Posselt et al. (2022). It consists of a free-running simulation across four two-way nested domains using version 3.9.1 of the WRF Model. The initiation of this simulation occurs at 0000 UTC on August 23, 2017, utilizing the initial state that generated the third most powerful member within an ensemble forecast of Hurricane Harvey. The ensemble's initial conditions were established through the assimilation of a conventional set of observations and all-sky satellite brightness temperatures. The NR simulation spans 5 days, ending at 0000 UTC on August 28, 2017, while the outermost domain's boundaries are guided by analysis fields from the fifth-generation European Centre for Medium-Range Weather Forecasts (ECMWF) Reanalysis (ERA5).

This simulation is notably realistic, capturing both wind patterns and humidity levels. Posselt et al. (2022) noted that it exhibits rapid intensification; within a span of 24 hours from 1200 UTC on August 24 to 1200 UTC on August 25, the minimum sea level pressure plunges by approximately 0.40 hPa, and the storm's strength escalates from category 1 to category 4. To get a view of different stages of Hurricane Harvey, we shall focus on two 12-hour subsets of the storm: one between 1800 UTC on August 23 to 0600 UTC on August 24 (Early Development Stage; Harvey EDS), and one between 0600 UTC on August 24 to 1800 UTC on August 24 (Late Development Stage; Harvey LDS). Note that with the division of Harvey into early and late development stages, we have a set of four scenarios– ETC, TC, and Harvey EDS and LDS – on which we shall focus our analysis.

## 2.2 Optical flow AMVs

In our effort to obtain accurate AMVs, we applied the advanced optical flow algorithm to four distinct simulated NatureRun datasets (for details, see Yanovsky et al., 2023). These datasets represented ETC, TC, and both the early and late stages of Hurricane Harvey. The analysis in (for details, see Yanovsky et al., 2023) revealed that the optical flow method had a distinct advantage over the traditional feature matching technique.

For every pair of images, the optical flow algorithm generated dense Atmospheric Motion Vectors for every pixel. On the other hand, the feature matching algorithm had its limitations – it was unable to generate AMVs in specific areas, particularly near domain boundaries. Although the optical flow approach did not perfectly capture the strong winds around the hurricane with absolute precision, the flow fields it produced closely resembled the wind fields observed in the natural runs datasets. A notable metric, the Root Mean Square Vector Difference (RMSVD), indicated that the errors associated with the optical flow algorithm were significantly reduced compared to those obtained with the feature matching algorithm. This resulted in an average accuracy improvement of about 30% to 50% for the four dataset analysed.

An important quality of the optical flow was its robustness. The results it produced remained relatively consistent, irrespective of the time interval. This was not the case with the feature matching method, whose results showed a significant change based on time intervals (Yanovsky et al., 2023). Given these advantages, we favor the optical flow as the preferred algorithm for retrieving AMVs in this OSSE exercise.

In Figure 1, we display the quiver plot of the wind vectors from optical flow (left panels) and NatureRun (middle column). The time stamp is chosen as the middle of the model run for each study scenario. The differences between the two wind



**Figure 1.** Quiver plot of the optical flow wind (left column), NatureRun wind (middle column) and differences (right column). The quivers are overlaid over a Yellow-Green heatmap of the water vapor, where yellow corresponds to low water vapor, and green corresponds to high water vapor. The rows correspond to ETC, MCS, Harvey EDS, and Harvey LDS. Note that the last column (wind differences) has been magnified in scale relative to the left and middle columns because we want to highlight the regions of significant differences.





fields (optical flow and NatureRun) are displayed in *magnified* scales in the right panels of Figure 1. We observe that the wind differences show a consistent pattern influenced by complex local factors, further complicated by what appears to be random variability and potential covariates such as wind rotation or water vapor gradients. Given our objective to model these error characteristics, we will approach the problem by first considering the space of predictor variables, often referred to as feature selection or variable selection.

## 3 Modelling Approach

### 3.1 Variable Selection


Before assessing the benefits of colocating passive and active wind data, we need focus on the issue of variable selection, which involves the identification of important variables or features for predicting the target quantity. In this context, our target is the bias between retrieved AMVs and actual wind values. This selection process holds significance due to its potential to

trim down the input parameter space. This reduction not only speeds up the training process but also enhances the model's reliability when dealing with unfamiliar data. Additionally, it contributes to simplifying the interpretation of model parameters.

Lidar instruments typical observe only one component of winds. Aeolus, for instance, measures "[the] component of the wind vector along the instrument's line of sight [HLOS]" (Lux et al., 2020). Here, we will similarly assume that the active instrument in our OSSE study will also observe only one component of the wind vector, though we will simplify the geometry

by assuming that simulated observable is the u-wind. This assumption is fairly benign, since it is simply a change of basis from the wind vector given by the HLOS wind and its (unobserved) perpendicular component to the much simpler $(u, v)$ basis.

Since we wish to model the bias between the optical flow $\hat{u}$ and the lidar $u$, we define the response variable as $y = \hat{u} - u$. As for the predictor variables, Posselt et al. (2019) examined the relationship between 'tracked' and 'true' wind using an OSSE framework for the same ETC region as this study, and they noted that there is considerable heteroskedasticity in the error

residual as a function of moisture, wind speed, moisture gradient, and wind-moisture gradient angle. Therefore, we shall start our list of potential variables as these four parameters. Since wind speed is simply a magnitude of the wind vector $(\hat{u}, \hat{v})$ in polar coordinates, we added the other component – wind angle – as well.

Further, we take advantage of the smooth output space of the optical flow algorithm to compute the first derivatives of $(\hat{u}, \hat{v})$, giving rise to a four dimensional vector $(d\hat{u}/dx, d\hat{u}/dy, d\hat{v}/dx, d\hat{v}/dy)$. These first derivatives are computed via first-order

finite differencing method. In theory, we could have also computed the second-order derivatives in this manner, but we opted otherwise due numerical instabilities that can result from computing high-order derivatives using finite differencing. From these derivatives, we added the curl and divergence to the list of potential variables. There variables are meant to inform the model of the rotation and flux of the wind field at any particular location. Further, we also computed the angle of the gradient, which is defined as the angle made with the x-axis by a 2-dimensional vector $(d\hat{u}/dx, d\hat{u}/dy)$ and $(d\hat{v}/dx, d\hat{v}/dy)$, respectively.

To generate the data for assessing the variable importance, we start with the arrays of optical flow u-and and v-wind, along with the water vapor content. We apply the finite differencing method to water vapor, u-wind and v-wind to generate the first-order derivatives, which then provides all the precursors necessary to compute the rest of the augmented variables described





**Figure 2.** Variable importance plots for the 500 hPa pressure level at ETC, MCS, Harvey2318, and Harvey2406 using three different approaches: random forest (left column), gradient boosting (middle column), and permutation with neural network (right column). Higher values indicate more importance.





| Name | Description |
|---|---|
| uopt | u-wind from optical flow ($\hat{u}$) |
| vopt | v-wind from optical flow ($\hat{v}$) |
| wind_angle | angle of the vector $(\hat{u}, \hat{v})$ with respect to latitudinal lines |
| wind_speed | $\sqrt{(\hat{u}^2 + \hat{v}^2)}$ |
| qv_gradient_1 | $dq/dx$ |
| qv_gradient_2 | $dq/dy$ |
| qv_gradient | $\sqrt{((dq/dx)^2 + (dq/dy)^2)}$ |
| QVangle | wind-moisture gradient angle |
| curl | curl of $(\hat{u}, \hat{v})$ |
| div | divergence of $(\hat{u}, \hat{v})$ |
| u_grad_angle | arctan( $d\hat{u}/dx, d\hat{u}/dy$ ) |
| v_grad_angle | arctan( $d\hat{v}/dx, d\hat{v}/dy$ ) |
| qv | water vapor content ($q$) |

**Table 2.** Names of variables used for variable importance analysis and their definitions.

above. At this point, each pixel in the domain can be represented by a 13-dimensional predictor vector (see Table 2 for a detailed description) and a scalar-valued response $y = \hat{u} - u$. We then converted this into a tabular format by randomly and

uniformly sampling 1% of the available domain for each time step and append them into a training-validation dataset. These datasets, which are in tabular format, then form the basis of the following error characterization exercises.

Having constructed the datasets, we consider the topic of variable importance. There is a large body of literature on the topic, particularly for regression-based methods. Examples include approaches such as genetic algorithms, jack-knifing, and forward selection (Bies et al., 2006; Lee et al., 2012; Blanchet et al., 2008). Here, due to the complexity of the functional model, we

select our variable set using three different machine learning approaches that have been shown to be capable of modelling highly multivariate functional relationships: random forest (Breiman, 2001), gradient boosting regression trees (Prettenhofer and Louppe, 2014), and multi-layer perceptron (Gardner and Dorling, 1998). (Further details of parameter optimizations for these methods are discussed in Section 3.2 and Table 3).

Random forest and gradient boosting, in this case, employ decision trees (Kingsford and Salzberg, 2008), which is a popular

and widely used machine learning algorithm that can be applied to both classification and regression tasks. Decision trees make predictions by mapping input features to output targets based on a series of binary decisions, and they form the basis of the two techniques considered in this section: random forest and gradient boosting trees. The metric for variable importance for these two methods are constructed by keeping track the decrease in accuracy or increase in impurity (e.g., Gini impurity for classification, increase in node purity for regression) caused by a chosen specific feature (Breiman, 2001). These purity-based





variable importances, where higher values indicate greater importance, are shown in the first and second columns of Fig. 2 for random forest and gradient boosting trees, respectively. The variables names on the x-axis are described in Table 2.

The results from the left and middle columns of Figure 2 indicate that the top five variables for regression are the retrieved optical flow winds $(\hat{u}, \hat{v})$, wind speed and angle, and water vapor (q). We note that wind speed and wind angle are the polar-coordinate transform of the rectangular coordinates $(\hat{u}, \hat{v})$, but their inclusion in the model significantly improves the model, since they provide an informative transformation that makes it easier for the machine learning model to model the functional form of interest.

One of the weaknesses of the purity-based variable importance plot is that high-correlation between features can inflate the importance of numerical features (Gregorutti et al., 2017; Nicodemus et al., 2010), and that the purity-based variable importance are based only on training data and can have low or no correlation with independent validation data. To address these short comings, we supplement them with another approach based on permutation, which could be applied any fitted estimator in tabular data contexts. The concept behind permutation feature importance involves quantifying the reduction in a model's score resulting from the random shuffling of a chosen variable (e.g., wind speed) while keeping all other variables the same within a fitted model (Breiman, 2001). The key insight is that if a model has significantly worse performance with a particular variable 'shuffled', then that variable must be important, and the degree of importance can be assessed by the magnitude of the performance degradation. One advantage of this technique is that it could be applied to non-linear or opaque estimators, and for this we choose to apply it in conjunction with a neural network, specifically a Multi-layer Perceptron regressor.

These permutation-based variable importance values are plotted in the right column of Figure 2. The variable importance that comes out of the permutation method has a different unit and scaling compared to purity-based variable importance but both are consistent in indicating that higher values signify greater importance. The overall patterns are the same between different approaches, indicating that for the most part, the most important variables are $\hat{u}, \hat{v}$), wind angle, wind speed, and water vapor content (q). Winds speed, however, is considered one of the most important predictors of bias according to the permutation method, and one additional feature that is considered somewhat important in this metric is the curl. Therefore, we shall consider the set of these six variables in the following analysis and modelling.

## 3.2 Algorithm comparisons

Having identified the important predictive variables, we consider the algorithms for fitting the bias functional form. The features we require of the algorithm are: able to handle complex multivariate data patterns, robust against new datasets, and computationally fast. For this reason, we have chosen four methods that are known to do well for high-dimensional problems with complex relationships: random forest, gradient boosting trees, multi-level perceptron, and nearest neighbor. Here, we touch briefly on an overview of the methods, before going into details of optimization and comparison.

Random Forest is a powerful ensemble learning technique used for both classification and regression tasks in machine learning. As the name suggests, it consists of an ensemble of multiple decision trees, combining these trees to create a more accurate and robust predictive model (Breiman, 2001). Random Forests are particularly popular due to their ability to handle





|  | Package Name | Parameter Settings |
|---|---|---|
| Random Forest | sklearn.ensemble.RandomForestRegressor | n_estimators $\in$ (100, 300, 500 ), criterion $\in$ ('squared_error', friedman_mse'), min_samples_split $\in$ (2,5,10), max_features $\in$ ( 'sqrt', 'auto' ) |
| Gradient Boosting Trees | sklearn.ensemble.GradientBoostingRegressor | learning_rate = (.01, .1, 1), n_estimators $\in$ (100, 300, 500 ), max_depth $\in$ (2, 4, 6), max_features $\in$ ( 'sqrt', 'log2' ) |
| Multi-Layer Perceptron | sklearn.neural_network.MLPClassifier | hidden_layer_sizes $\in$ ( [30, 15], [20, 20], [30, 20], [20,10] ), activation $\in$ ( 'tanh', 'logistic', 'relu' ), learning_rate $\in$ = ('constant', 'invscaling', 'adaptive'), max_iter = (200, 500, 1000 ), learning_rate_init $\in$ (1e-3, 1e-4, 1e-5) |
| Nearest Neighbor | sklearn.neighbors.NearestNeighbors | algorithm = 'KDTree', n_neighbors $\in$ (3, 5, 10), p $\in$ (1, 1.5, 2 ) |

**Table 3.** Python package names (middle column) and the parameter settings for each of the method. Parameters not mentioned on this table are set as default in the python methods.

complex data, reduce overfitting, and provide valuable insights into feature importance. Each tree is constructed using a random
subset of the training data and a random subset of the input features. The predicted value, whether a class label or a regression value, is computed by passing the predictors to all the trees fitted within the model and aggregating the corresponding outcomes.

Gradient Boosting Trees is another powerful machine learning technique falling under the ensemble methods umbrella. Like random forests, Gradient Boosting constructs an ensemble of decision trees, with each referred to as a 'weak learner' because it is relatively simple and typically underfits on its own. However, the trees are built sequentially, with each new tree aiming
to correct the errors made by the previous ones. Similar to random forests, Gradient Boosting aims to build many trees and is widely used for both regression and classification tasks because of its capacity to create accurate predictive models capable of handling complex data patterns (Prettenhofer and Louppe, 2014).

The Multi-Layer Perceptron (MLP) is a foundational artificial neural network architecture that serves as the cornerstone for deep learning models. It is a versatile and powerful technique used for a wide range of machine learning tasks, including
classification and regression. An MLP consists of interconnected layers of artificial neurons or nodes, roughly divided into three types: input layers, which typically represent the predictors; hidden layers, responsible for processing information from the previous layer and extracting relevant features; and the output layer, which produces the final result, such as a classification label or a regression value. Each neuron processes information and passes its output to the next layer, and the numeric parameters within each node, namely the weight and bias values, are estimated from the data using backpropagation and gradient descent
(Gardner and Dorling, 1998).





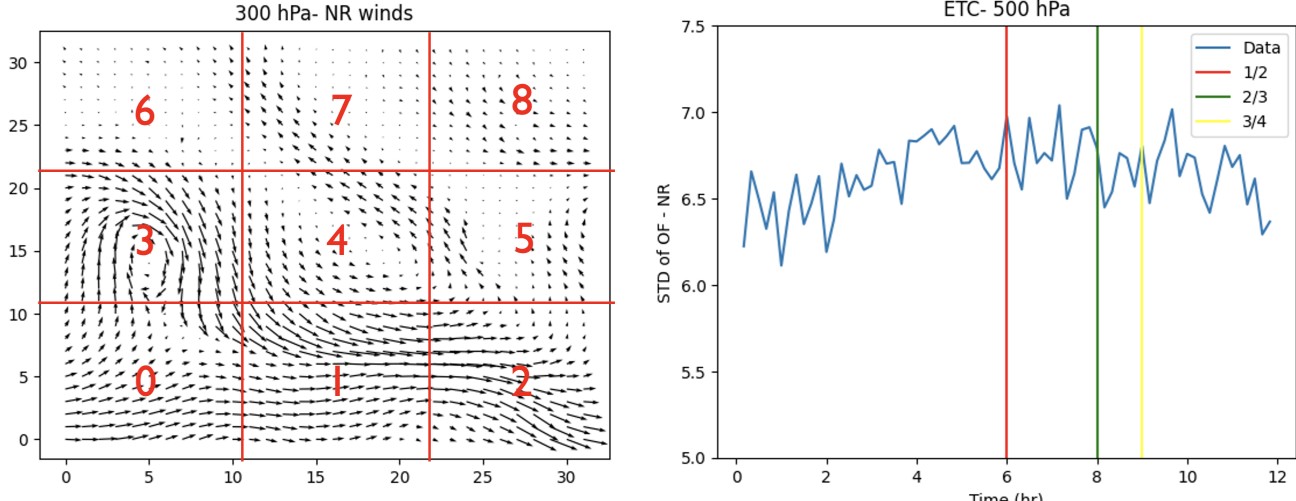

**Figure 3.** Left panel: spatial validation scheme where the domain is divided into a 3x3 grid and labelled from 0 to 8. Right panel: temporal validation scheme where the training data is set as the first 1/2, 2/3, and 3/4 of the full dataset, respectively.

Nearest neighbor methods operate on the principle of identifying a fixed number of training samples that are closest in proximity to the new data point, and then predicting the label based on these identified neighbors. This number of samples can either be a user-defined constant, characteristic of k-nearest neighbor learning, or it can adapt based on the density of nearby points, as seen in radius-based neighbor learning. The measurement of distance can be achieved through various metric measures, with the standard Euclidean distance being the most commonly selected option.

We used implementations of these methods from the Python scikit-learn package (version 1.2) (Kramer and Kramer, 2016). All of these methods require tuning of algorithm parameters, such as tree leaf nodes and depth for random forest and gradient boosting, hidden layer sizes, and activation methods for the neural network, neighbor size, and distance metrics for nearest neighbors, etc. To optimize these parameters, we employed the Grid Search optimization method from the scikit-learn package (sklearn.model_selection.GridSearchCV). This method iterates through different parameter choices provided in the parameter grid and identifies the best combination of parameters that minimize the loss function, which in this case is the root mean-squared error when fitted against the training data. The parameter search space for these four methods is detailed in Table 3.

To evaluate the performance of the four different methods, we divided the tabular datasets created from the data arrays into training datasets (used for model building) and validation datasets (used for performance assessment). We employed two types of division - spatial and temporal - as illustrated in Figure 3. In the temporal division, we reserved the last 1/4, 1/3, and 1/2 of the data using timestamps, respectively, and utilized these withheld data to evaluate performance in terms of RMSE. For the ETC dataset, spanning 12 hours, this entailed setting aside the last 3, 4, and 6 hours of data for validation.





|  | RandomF | GradientB | NeuralN | NearestN | MarginalSTD |
|---|---|---|---|---|---|
| ETC- 300 hPa | -0.596 | -0.680 | -1.079 | -0.707 | -3.233 |
| ETC- 500 hPa | -0.1030 | -0.294 | 0.128 | -0.217 | -2.240 |
| ETC- 850 hPa | -0.017 | -0.053 | 0.006 | -0.096 | -0.477 |
| TC- 300 hPa | -0.240 | -0.206 | -0.408 | -0.268 | 0.806 |
| TC- 500 hPa | 0.072 | 0.100 | 0.003 | 0.130 | 0.423 |
| TC- 850 hPa | -0.071 | -0.087 | -0.191 | -0.074 | 0.079 |
| Harvey EDS- 300 hPa | 0.367 | 0.385 | 0.344 | 0.428 | -0.160 |
| Harvey EDS- 500 hPa | -0.049 | -0.043 | 0.020 | -0.043 | -0.362 |
| Harvey EDS- 850 hPa | 0.061 | 0.020 | 0.075 | 0.056 | 0.131 |
| Harvey LDS- 300 hPa | 0.182 | -0.027 | 0.128 | 0.065 | -0.861 |
| Harvey LDS- 500 hPa | 0.344 | 0.338 | 0.409 | 0.233 | 0.219 |
| Harvey LDS- 850 hPa | -0.272 | -0.163 | -0.179 | -0.210 | -0.095 |

**Table 4.** Validation temporal bias (computed from withheld last half of data for each atmospheric regime) for Random Forest, Gradient Boosting, Neural Network, and Nearest Neighbor. Units are in m/s.

The results of this temporal validation for the ETC case are displayed in the right panel of Figure 4. In all pressure levels,
the machine learning approach consistently exhibits smaller bias than the uncorrected optical flow data. Notably, for the 300
and 500 hPa levels, the bias magnitude is significant at 2.5-3 m/s, but it is reduced to less than 0.5 m/s, signifying a substantial
improvement. Similarly, although the reduction in bias is smaller at 850 hPa due to the optical bias starting at a lower value,
the same trend persists.

It is informative to assess the performance of all four algorithms across the four scenarios and three pressure levels. There-
fore, we selected the case where 1/2 of the data was withheld (considered the most challenging case) and summarized the
validation performance in terms of RMSE for all four methods in Table 4. The scenarios at different pressure levels are listed
in the rows. Overall, random forest, gradient boosting, and MLP tend to exhibit comparable performance, with no clear pref-
erence among the three. Nearest neighbor, on the other hand, consistently reduces bias relative to the uncorrected optical flow
but falls short of the performance achieved by state-of-the-art ensemble methods and neural networks. This suggests that the
proximity-based methodology may not be flexible enough to capture the complex dependence structure of the AMV biases.
We note that there are three instances where the uncorrected optical flow has the smallest bias (Harvey EDS 300 hPa, Harvey
LDS 500, Harvey LDS 850), but these cases share a common feature in which the original optical flow exhibits very low bias
(<0.25 m/s). In such cases, the algorithms may struggle due to the limited discernable signal for modeling.

Another validation approach is a purely spatial one. In this approach, we divide the domain of each study region (as seen in
Figure 1) into equal 3x3 areas and label each subregion with an index ranging from 0 to 8. In this labeling scheme, 0 represents
the bottom-left cell, 2 the bottom-right cell, 4 the center cell, and 8 the top-right cell. We then withhold one of these nine





**Figure 4.** Sample plot of the random forest performance for ETC at three pressure levels for the spatial case (left column) and temporal case (right column).





| | RandomF | GradientB | NeuralN | NearestN | Uncorrected |
|---|---|---|---|---|---|
| ETC- 300 hPa | 2.875 | 2.890 | 2.711 | 3.257 | 4.048 |
| ETC- 500 hPa | 1.832 | 1.817 | 1.836 | 2.142 | 2.688 |
| ETC- 850 hPa | 0.695 | 0.685 | 0.801 | 0.743 | 0.749 |
| TC- 300 hPa | 1.403 | 1.276 | 1.234 | 1.3417 | 1.139 |
| TC- 500 hPa | 0.546 | 0.581 | 0.728 | 0.610 | 0.648 |
| TC- 850 hPa | 0.493 | 0.489 | 0.442 | 0.468 | 0.463 |
| Harvey EDS- 300 hPa | 0.430 | 0.379 | 0.477 | 0.437 | 0.455 |
| Harvey EDS- 500 hPa | 0.661 | 0.556 | 0.765 | 0.691 | 0.485 |
| Harvey EDS- 850 hPa | 0.435 | 0.318 | 0.483 | 0.477 | 0.379 |
| Harvey LDS- 300 hPa | 0.688 | 0.732 | 0.936 | 0.825 | 0.936 |
| Harvey LDS- 500 hPa | 0.240 | 0.344 | 0.333 | 0.377 | 0.376 |
| Harvey LDS- 850 hPa | 0.566 | 0.499 | 0.379 | 0.601 | 0.372 |

**Table 5.** Validation spatial bias (averaged over all 9 spatial regions) for Random Forest, Gradient Boosting, Neural Network, and Nearest Neighbor. Bias are computed as the mean of absolute bias within each of the spatial region.

regions at a time and train our models (e.g., random forest, gradient boosting, etc.) on the other eight cells. Subsequently, we apply our trained model to the withheld region. A sample of the results from these spatial validation efforts is shown in the left panel of Figure 4 for the ETC case, using the random forest algorithm. In this figure, the indices on the axes represent the region that was withheld from the training process. The overall biases, computed in the lower right of the panels, are mean absolute biases (MABs), which are defined as

$$\text{MAB(m)} = \sum_{i=1}^{9} \frac{|b_i(m)|}{9} \text{ where } b_i(m) = \sum_{j=1}^{N_i} \frac{(\hat{u}_{ij}^m - u_{ij})}{N_i}.$$

In this formula, $m$ represents the methodology being evaluated (e.g., random forest, gradient boosting, optical flow, etc.), $b_i(m)$ denotes the normal bias when applying methodology $m$ to the $i$-th withheld dataset, and $N_i$ stands for the number of observations within the $i$-th validation dataset. The reason for calculating the mean absolute biases across these nine regions is to account for the possibility of biases having both positive and negative values. Therefore, we take the absolute value before averaging to prevent negative biases from canceling out positive biases and potentially distorting the resulting metrics.

The results from the ETC case in Figure 4 indicate that, for most of the regions, the trained model results in biases that are smaller in magnitude than those of the original optical flow u-wind. In some cases, the improvement in bias can be substantial (e.g., region 1). While in a few instances, the RF model can result in biases with increased magnitude, this adverse effect is generally small compared to the magnitude of gains observed in other regions. The Mean Absolute Biases (MABs) are displayed in the lower right corner of the left panels in Figure 4, and they suggest that random forest consistently reduces the magnitude of the bias compared to the optical flow data.





In Table 5, we present the MABs of the four algorithms (as well as that of the uncorrected optical flow) across the four

scenarios and pressure levels. We observe the same overall patterns as in the temporal validation shown in Table 4, noting that random forest, gradient boosting, and MLP tend to exhibit the best performance, although their dominance varies across different scenarios and pressure levels. As before, nearest neighbor does not produce notably superior results. In some cases, the uncorrected optical flow algorithm has the lowest error, but these tend to be cases where the bias initially starts off at a low level.

## 3.3   Uncertainty characterization of AMVs

In data assimilation, thinning high-density AMVs is often necessary. Typically, this process involves giving preference to vectors that exhibit higher accuracy. The selection procedure usually takes into consideration various indicators of error level associated with the vectors, such as the quality indicator (QI), expected error (EE), recursive filter flag (RFF), error flag (ERR), and others (Le Marshall et al., 2004). All of these metrics share the common goal of grouping AMVs with similar errors

together, meaning that observations with good quality indicators should all exhibit low errors relative to the unobserved truth.

**Table 6.** Multi-row table

|  |  | High Quality (STD) | Mid Quality (STD) | Low Quality (STD) | Marginal Data |
|---|---|---|---|---|---|
| ETC | 300 hPa | 3.510 | 3.928 | 4.887 | 4.180 |
|  | 500 hPa | 2.550 | 3.364 | 4.738 | 3.634 |
|  | 850 hPa | 1.808 | 2.541 | 3.706 | 2.784 |
| TC | 300 hPa | 1.232 | 1.962 | 3.110 | 2.190 |
|  | 500 hPa | 1.291 | 1.659 | 2.170 | 1.788 |
|  | 850 hPa | 1.231 | 1.537 | 1.864 | 1.568 |
| Harvey EDS | 300 hPa | 0.910 | 1.382 | 2.391 | 1.716 |
|  | 500 hPa | 1.093 | 1.451 | 2.395 | 1.775 |
|  | 850 hPa | 1.187 | 1.476 | 2.033 | 1.623 |
| Harvey LDS | 300 hPa | 1.293 | 2.320 | 3.556 | 2.536 |
|  | 500 hPa | 1.260 | 1.756 | 2.701 | 2.021 |
|  | 850 hPa | 1.450 | 1.963 | 2.579 | 2.073 |

**Table 7.** STD vs Quality Indicators based on RF prediction errors

We note that pattern tracking and optical flow do not provide an intrinsic error estimate, necessitating the addition of the post-hoc error indicators mentioned above. In the existing remote sensing literature, a variant of random forest called quantile random forest has been extensively used to model uncertainty alongside the prediction of interest (e.g., digital soil mapping product, Vaysse and Lagacherie (2017); soil organic matter, Nikou and Tziachris (2022); nitrogen use efficiency, Liu et al.



**Figure 5.** Actual coverage percentage of the 95% prediction intervals when evaluated against withheld validation data. The vertical red line is the ideal coverage percentage as implied by the 95% intervals.

(2023)). In this section, we shall employ quantile forest regression to construct the prediction intervals for the u-wind and compare them with withheld validation u-wind data.

Quantile random forest, as introduced by Meinshausen and Ridgeway (2006), is a modification to the random forest procedure that enables the estimation of prediction intervals for the intended variables. In contrast to normal random forests, which approximate the conditional mean of a response variable, quantile regression forests (QRF) provide the full conditional 320 distribution of the response variable to construct prediction intervals. The key insight that allows for this property is that, while random forests retain solely the mean of observations within each node and discard any additional information, quantile regression forests preserve the values of all observations within the node, not just their mean, and use these distributions to make estimates of the quantiles of interest. In particular, Meinshausen and Ridgeway (2006) prove that the conditional quantile estimates are asymptotically consistent under specific assumptions:



**Figure 6.** Plots of the estimated prediction error from random forest versus empirical validation error using quantile regression for ETC (top left), TC (top right), Harvey EDS (bottom left), and Harvey LDS (bottom right). Red lines are the identity ($y = x$) lines.





1. The proportion of values in a leaf, relative to all values, vanishes as the number of observations $n$ approaches infinity.

2. The minimal number of values in a tree node grows as $n$ approaches infinity.

3. When looking for features at a split, the probability of a feature being chosen is uniformly bounded from below.

4. There is a constant $\gamma$ in the range $\gamma \in (0, 0.5)$ such that the number of values in a child node is always at least $\gamma$ times the number of values in the parent node.

5. The conditional distribution function is Lipschitz continuous with positive density.

These are fairly modest assumptions, particularly with respect to the construction of the trees. However, it's worth noting that the quantiles under these assumptions is only asymptotic consistent as $n$ approaches infinity. However, these assumptions provide some theoretical assurance that the outputs of quantile random forest should approximate the true conditional quantiles to some extent.

Here we use the python implementation of QRF provided by the Python package *quantile-forest*. We use quantile regression forest to build 95% prediction intervals at the pixel level. That is, at any pixel, we compute the prediction interval as follows

$$I(\boldsymbol{x}) = [Q_{.025}(\boldsymbol{x}), Q_{.975}(\boldsymbol{x})]$$

where $Q_{.025}(\cdot)$ and $Q_{.975}(\cdot)$ are the 2.5-th percentile and 97.5-th percentile random forest estimators from *quantile-forest*, respectively, and $\boldsymbol{x}$ is the vector of predictors (e.g., optical flow winds, wind speed, angle, etc.) as discussed in Section 3.1.

We wish to assess the performance of the intervals $I(\boldsymbol{x})$ given by quantile random forest, and one approach is to compute the coverage probability of the confidence intervals when applied to withheld validation data. We repeat the exercises in the previous section, and for each of the four scenarios and three pressure levels, we use the first half of the storm for training, and the second half for validation. We then compute the coverage percentage of the prediction intervals, which is defined as the probability (expressed as percentage) that the true u-wind actually falls within the interval given by quantile regression. That is, the coverage percentage (CP) for a given scenario and pressure level is given by

$$CP = \frac{\sum_{i=1}^{N} I\left(Q_{.025}(\boldsymbol{x}_i) \leq u_i \cap u_i \leq Q_{.975}(\boldsymbol{x}_i)\right) \cdot 100}{N}$$

where $u_i$ is the $i$-th WRF wind from the withheld validation set, $I(\cdot)$ is the indicator function, and $N$ is the size of the validation dataset. A comparison of the coverage probability for all scenarios and pressure levels is given in Figure 5. There, we see that the 95% prediction intervals from quantile random forest consistently *underestimate* the magnitude of the error variability, averaging between 80-85% coverage while the ideal number should have been 95%. This implies that the prediction interval widths given by quantile random forest in general tend to be a bit smaller than what the validation data require.

To get a clearer idea of the differences between the estimated prediction error and that of the validation data, we examine their relationship in a scatter plot. To do so, we first convert the prediction intervals $I(\boldsymbol{x})$ to their effective prediction error $\hat{\sigma}(\boldsymbol{x})$. This conversion relies on the fact that in a Gaussian distribution, the 95% confidence interval is given by a $+/- 2$ standard





deviation of the mean. Therefore, to compute the effective prediction error from our 95% confidence interval, we divide the interval width by 4. That is,

$$\hat{\sigma}(\boldsymbol{x}) = \frac{Q_{.975}(\boldsymbol{x}) - Q_{.025}(\boldsymbol{x})}{4}.$$

To compare these effective prediction error $\hat{\sigma}(\boldsymbol{x})$ against the withheld validation data, we construct the equivalent standard error using the *withheld validation data*. We use the binning approach, where the empirical validation error (EVE) for a given prediction error value $\sigma$ and a given bin length $d$ is computed by aggregating all observations where the QRF prediction error is within $+/-d$ of $\sigma$, and then we compute the RMSE on this subset. In formal terms, the EVE is given by:

$$EVE(\sigma, d) = \sqrt{\left( \sum_{i \in |\hat{\sigma}(\boldsymbol{x}_i) - \sigma| < d} \frac{(u_i^* - u_i)^2}{N} \right)} \tag{1}$$

where $u_i^*$ is the $i$-th RF-corrected u-wind in the validation data set, $u_i$ is the $i$-th true WRF u-wind, and $N$ is the size of the validation dataset.

With the formulas above, we binned the RF prediction errors into eight equally-spaced bins and computed the corresponding EVE. The results are shown for all scenarios at the 500 hPa pressure level in Figure 6. Although the overall patterns are similar at other pressure levels, this figure provides a more nuanced view, naming that while the RF prediction errors tend to under estimate the true errors, they exhibit a *monotonic increasing* relationship. This is a valuable property. It implies that while the errors are not accurate (i.e., statistically valid), their monotonic increasing correlation with the true error indicates that we can use the QRF prediction error as a proxy for quality assessment.

To demonstrate the usefulness of this property, we simulate a quality indicator flag by dividing the validation data into three equal-size categories: low, mid, and high quality. The three categories are constructed by sorting the QRF prediction errors (or alternatively the prediction interval widths) from smallest-to-largest, and then classifying the smallest one-third as high-quality, the middle one-third as mid-quality, and the largest one-third as low-quality. We then compute the EVE of the optical flow wind versus the withheld 'true' wind within each bin, and we display the results in Table 7. Intuitive understanding of high-quality observations generally implies that they are more 'accurate' than low-quality observations, and indeed here Table 7 indicates that the EVE values, for every single pressure level and region, form an increasing pattern with high-quality observations having the smallest error, mid-quality having medium error, and low-quality having the largest error. The differences between the high-quality and low-quality observations can be fairly significant, with many rows having high-quality EVE values that are more than 50% smaller than that of the low-quality observations.

We note that the experiment in Table 7 divided the data into three bins. In general, the results here should hold for different numbers of bins, although a high single-digit number might be unstable. A hint of this instability is seen in Figure 6, where we observe that for the lower-right panel, the right-most bin actually has a slightly smaller EVE than the bin immediately preceding it. This may be due to the fact that there are low bin counts at the extreme edge of the domain. In general, increasing the bin count can reduce the bin counts, exacerbating these statistical artifacts. However, quality indicators in common usage tend to use a low single-digit number of bins, which works well here.



### 3.3.1 QRF prediction intervals versus RMSVD

In the previous subsections, we have demonstrated that the combination of passive winds with active sensor u-wind can help improve the bias and uncertainty characterization of the u-component. However, error metrics of interest in winds, such as the
root mean squared vector difference (RMSVD), often encompass the full wind vector. The question is whether the promising property we observed in Section 3.3 remains when comparing the QRF error predictions to the full-vector RMSVD. To address this, we repeat the previous exercise. For each of the four scenarios and three pressure levels, we use the first half of the storm for training and the second half for validation. Instead of measuring the standard deviation with respect to the u-component, as in Equation 1, we calculate the validation error as the RMSVD, which is defined as:

$$
\quad RMSVD = \sqrt{\left( \sum_{i=1}^{N} (\hat{u}_i - u_i)^2 + (\hat{v}_i - v_i)^2 \right)}. \tag{2}
$$

The updated plots of QRF prediction error versus RMSVD are displayed in Figure 7 for the 500 hPa pressure levels. Notably, the primarily positive linear trends are still evident, though the relationship is no longer strictly monotonically increasing. This characteristic is promising, suggesting that, in general, having coincident lidar u-wind data with passive AMVs will allow for estimates of the RMSVD that are mostly accurate in a relative sense when comparing between pixels. Due to space constraints,
this paper only includes the plots for 500 hPa. However, similar relationships are observed for the other pressure levels. As in the previous section, this relationship between QRF error prediction and the RMSVD can be leveraged to create quality indicator flags, error flags, or filter flags.

While it is evident that uncertainty predictions based on u-wind from lidar contain information about the total-vector RMSVD, this property may not hold universally. A closer examination of the datasets reveals that this property arises from two
patterns: a) the uncertainty prediction increases with higher u-wind errors, and b) the RMSVD has a positive correlation with u-wind error. To illustrate this, we present these relationships in Figure 8.

In the top row, we can see that the average QRF prediction errors grow larger as the absolute u-wind difference ($|\hat{u}_i - u_i|$) increases. This makes sense intuitively, and it indicates that in general, pixels that are relatively unbiased (i.e., $|\hat{u}_i - u_i| \approx 0$) are relatively easy for the quantile random forest algorithm, which results in a relatively small prediction interval. As the magnitude
of the difference increases, the algorithm grows more and more uncertain of its prediction, and hence the prediction interval increases in width.

In addition to the positive relationship between QRF prediction width and u-wind error magnitude, we can also see in the second row of Figure 8 that the RMSVD also has a positive correlation with the magnitude of the u-wind error. This makes sense intuitively because $|\hat{u}_i - u_i|$ is one half of the definition of the RMSVD as expressed in Equation 2, and therefore it is
not surprising that the two exhibit positive correlation with one another.

Therefore, the relationship between u-wind QRF prediction intervals and RMSVD can be explained by the fact that higher QRF prediction intervals tend to occur when the actual error magnitude (i.e., $|\hat{u}_i - u_i|$) is larger, and that in turn implies that RMSVD is also higher due to the natural correlation between $|\hat{u}_i - u_i|$ and RMSVD. In this OSSE, we have made the simplifying assumption that the active instrument's observable is the $u$-wind. This same positive relationship should extend





**Figure 7.** Plots of the estimated prediction error from quantile random forest versus RMSVD for all four study scenarios. Red lines are the identity ($y = x$) lines.

to HLOS winds and RMSVD, since this is simply a change of basis for the wind-vector without impact to the underlying functional relationships.

## 4   Conclusions

Accurately estimating global wind patterns is of paramount importance across scientific and practical domains, including applications like global chemical transport modeling and numerical weather prediction. Atmospheric Motion Vectors (AMVs) serve as crucial inputs for these applications. However, addressing errors in AMV retrievals becomes imperative before their assimilation into data assimilation systems, as these errors can significantly impact output accuracy. One noteworthy error

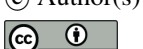

**Figure 8.** Plot of relationships between RF prediction error, absolute u-wind difference, and RMSVD for ETC (top row) and MCS (bottom row).





characteristic of AMVs is bias, which varies considerably by region. For instance, Staffell and Pfenninger (2016) found that NASA's MERRA and MERRA-2 AMVs tend to overestimate wind output by 50% in northwest Europe and underestimate it by 30% in the Mediterranean. These biases can lead to adverse results if the AMVs are incorporated into data assimilation
systems without proper mitigation or bias removal.

In real-world applications, correcting the bias in AMV retrievals necessitates an independent benchmark or reference to establish accuracy. Independent data sources may include collocated radiosonde data or lidar AMV data, such as those available from Aeolus. In this paper, we present a proof-of-concept that demonstrates the feasibility and performance of a bias-correction scheme within an Observing System Simulation Experiment (OSSE) framework. Specifically, we examined three different
storm systems in the Gulf of Mexico, North Atlantic Ocean, and Southeast Asia, and applied our bias correction and prediction error interval procedure to outputs generated by a novel AMV algorithm known as optical flow. Our results suggest that passive-sensor AMVs, which typically have high coverage but low precision, can benefit significantly from coincident high-precision active-sensor wind data. These benefits can be harnessed through algorithms that model expectations (bias reduction) or quantiles (uncertainty quantification).

In Section 3.2, we demonstrated that conventional machine learning algorithms such as random forest and gradient boosting can effectively learn the complex multivariate dependence structure of errors and correct biases in raw optical flow AMVs. It's worth noting that, despite having low bias in some cases, the standard deviation of the AMV error can be relatively large (e.g., with a standard deviation of 1-2 m/s while the error may be on the scale of <0.5 m/s). In these scenarios, the error-correction model produces biases of a similar magnitude (i.e., <0.5 m/s). Notably, we showed that, in the storm systems we considered,
it is possible to estimate biases with minimal performance degradation up to six hours in advance.

One of the most valuable extensions of machine learning models in this bias-correction exercise is the ability to estimate prediction intervals. In Section 3.3, we employed the quantile regression framework by Meinshausen and Ridgeway (2006) to generate prediction intervals for withheld validation data. We observed that, while the prediction intervals often tend to be too narrow (underestimating the variability of the true process), they generally exhibit a monotonically increasing relationship with
the true AMVs. In other words, the uncertainty estimates from the quantile random forest are not statistically valid (i.e., the 95% confidence intervals may not capture the truth 95% of the time), but the algorithm does correctly rank the error magnitudes when analyzing multiple pixels. Therefore, while the prediction intervals may not be directly usable in data assimilation, they can serve as valuable components of a quality indicator. Indeed, in Section 3.3, we conducted an experiment where we categorized the optical flow retrievals into three groups — high, mid, and low quality. We demonstrated that the standard
deviation within these categories relative to the validation data follows an increasing pattern, with high-quality observations having the lowest error standard deviation, mid-quality observations falling in the middle range, and low-quality observations displaying the highest error standard deviation.

It's important to highlight that our results primarily focus on errors related to a single wind component, as lidar systems typically only observe winds along the line-of-sight. In this OSSE study, we simplified the line-of-sight direction to align with
the u-wind direction, and we have shown that the uncertainty in the u-wind bias has a positive correlation with the full-vector RMSVD. This relationship arises from two characteristics of the wind errors in our study area. First, wind prediction errors



tend to increase with the absolute u-wind error. Second, the RMSVD demonstrates a positive correlation with the absolute u-wind error. We anticipate that these findings will generalize to the relationship between the HLOS wind and the full-vector wind in other regions, as this is essentially a change of basis for the $(u, v)$ wind components.

These results are highly promising, particularly regarding the application of quantile random forest in quality indicators. Our future research plans involve extending this study to other global regions and various convective systems. It's worth noting that different applications or study regions may necessitate distinct sets of predictive variables, and the selection of variables employed in our feature selection process may not be universally applicable. Nevertheless, the same variable selection process presented in Section 3.1 can be adapted to determine the most relevant predictive variables. In this paper, we utilized quantile

random forest for prediction interval estimation, but theoretically, other machine learning algorithms could be employed to generate quantiles (e.g., quantile neural networks, etc.), although the computational requirements may vary.

*Code and data availability.*   The data that support the findings of this study are not publicly available due to their large volume. The data and codes are available on request from the corresponding author.

*Author contributions.*   H. Nguyen designed the experiments, performed the computations, and wrote the paper. L. Wu generated the WRF
NaturenRun data, and I. Yanovsky performed optical flow retrievals. D. Posselt and S. Hristova-Veleva contributed to the design of the methodology and the validation exercise. All authors participated in providing feedback on the design and findings of the paper.

*Competing interests.*   The authors declare that they have no conflict of interest.



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
