# Peer review of "AMV Error Characterization and Bias Correction by Leveraging Independent Lidar Data: a Simulation using OSSE and Optical"

_Atmospheric Measurement Techniques, 2023_

## Referee Comment (RC1)

**Review - AMV Error Characterization and Bias Correction by Leveraging Independent Lidar Data: a Simulation using OSSE and Optical Flow AMVs**

Overall the proposed scheme is interesting and a useful contribution to the field. Up to and including the bias correction scheme it is well described. After that once we get into (I think) a proposed error estimation scheme I found it difficult to understand. I've picked out a few things that were unclear to me, if these things are explained in the text I'm happy to be corrected.

Quality of English is good, it would benefit from another look through by the authors to fix a few grammatical errors. Examples (not an exhaustive list):
Line 51 an bias-correction should be a bias correction
Line 88 AMV should be AMVs
Line 171: due -> due to
Line 193:  track -> track of

Comments by section:

Section 2.1

In section 2.1 it would be worth explaining how the heights were chosen – traditional AMVs are concentrated around 850 and 200 hPa, so 300 hPa is a little low in the atmosphere and there are normally few quality AMVs at 500 hPa.

In the same section, line 115-6, why only conational obs + brightness temperatures to initialise, and not all observation types? And on line 114-5, why the 3$^{rd}$ most powerful member?

Line 121: I don't know a great deal about tropical cyclones but a drop of 0.4 hPa seems very small and not a 'plunge'

Section 2.2:
Although you refer to another paper, you could state here which traditional feature tracking algorithm you're referring to and AMV people will know its main strengths and weaknesses.

Line 128: what do you mean by dense AMVs? If they are derived for every pixel that already implies they are spatially dense, so are they dense in some other way?

What part of the NatureRun is used to derive the AMVs – the clouds, as in traditional AMV feature matching? Or the humidity surfaces, or something else?

The main source of error in AMVs is generally thought to be height assignment. If the optical flow is done on WRF humidity fields, then the error is only coming from the tracking and likely to be smaller than typical AMV errors

On figure 1 a scale would be helpful, to know how fast the winds are, and how large the differences are. The factor by which the differences are magnified should also be stated.

Section 3.1

I was a bit confused here, are we treating the WRF winds as truth and trying to bias correct towards WRF? Or are we simulating some Aeolus-style u-winds from WRF and bias correcting towards them? How are the lidar winds simulated? There should be some error associated with such lidar winds.

What is meant by wind-moisture gradient? Also, I've never heard the word heteroskedasticity and had to look it up, could you define it in the text or explain what you mean without using it? What is 'error residual'?

Figure 2 caption – mention that the labels are defined in Table 2

Table 3 caption, mention that the numbers in arrays are being searched through.

Section 3.2

Line 238-245: I appreciate that there is an attempt to explain MLP but there are lots of new terms that are not fully explained for example nodes and neurons, backpropagation, gradient descent

Line 250: what is the standard Euclidean distance?

Figure 3 – what is NR?

Table 4 – what is MarginalSTD and why are some numbers red?

Lines ~264-281: by this point we're talking about the bias of the optical flow winds (with or without bias correction applied) vs the WRF simulation truth? Is that right? Perhaps worth re-mentioning the first time 'bias' is mentioned.

Line 274: what is meant by state-of-the-art ensemble methods and neural networks? Do any of the methods being tested fit this description?

Table 5 – perhaps I missed this in the text, but why are they biases so much larger with the spatial validation than with the temporal validation

Section 3.3

Table 6 – looks like a caption for a non existent table

Table 7: does it say in the text somewhere which data is Marginal?

Normally, AMV quality indicators are given as an integer from 0 to 100. Why is a different scale used for these quality indicators?

Page 19: I found this page quite a struggle, I get that you are trying to come up with error estimates, but what are prediction intervals, nodes, 'leaf' etc? Is QRF quantile random forest or quantile regression forest?

Table 7 is quite a few pages away from where it is referred to, it would be good to move it closer.

Section 3.3.1

Line 398: Here it is suggested that the error estimation would only work when lidar data is coincident. I though the idea was to train a model using lidar data, that can then be used any time, anywhere? If the proposed scheme would only work when, for example, Aeolus data happens to be in the same place as AMVs then the scheme would only apply to a very small fraction of AMVs were it used operationally.

Section 4 (Conclusions)

Line 450: true AMVs = WRF NatureRun, correct? Better to say so

---

## Referee Comment (RC2)

**Referee Comments**

Manuscript: amt-2023-239

**Title: AMV Error Characterization and Bias Correction by Leveraging Independent Lidar Data: a Simulation using OSSE and Optical Flow AMVs**

Author(s): **Hai Nguyen et al.**

**General Comments:**

The manuscript explores the application of machine learning techniques to assess bias and uncertainty in the assimilation of atmospheric motion vectors (AMVs). The authors frame the problem by treating independent LIDAR wind observations as a dependent variable in a supervised learning machine model. The study utilizes an Observing System Simulation Experiment (OSSE) framework, with reference geophysical state data derived from high-resolution Weather Research and Forecasting (WRF) simulations.

The literature review is comprehensive, providing a strong foundation for the study. The motivation for the research is clearly articulated. However, it's crucial to note that the paper primarily serves as a proof-of-concept, a fact that becomes evident through the text. While the title implies a broader scope, the content remains focused on the proposed machine learning approach for bias correction in wind field assimilation.

The approach presented is sound, addressing and resolving issues identified in previous methodologies. The paper is well-structured, and the visual aids effectively support the discussion. However, there are opportunities to better depict certain concepts, as outlined below.

**Specific Comments:**

1. The authors should provide a more detailed explanation of their efforts to obtain accurate AMVs. Although they refer readers to another publication for details, as that reference is still "submitted for publication," a general explanation or summary is important for proper understanding.

2. Clarity regarding the connection between the proof-of-concept and the utilization of Lidar data is essential. It seems that certain errors associated with Lidar wind profiles were not considered, impacting the comprehensiveness of the study. Clarifying this aspect would strengthen the paper.

3. The presentation of optical flow could be improved for better interpretation. The right column in Figure 1, in particular, may benefit from replacing arrows indicating differences with a color-coded scale. Additionally, consider addressing potential confusion related to the arrows' direction by emphasizing differences in magnitude rather than implying directional changes.

4. While the paper is technically sound, providing a more explicit link between the proposed methodology and Lidar data considerations would enhance the manuscript's overall coherence and contribute to a more comprehensive understanding for readers.

**Technical Corrections:**

| | |
|---|---|
| The first abbreviation of Observing System Simulation Experiments should appear in **Line 49.** | *"Observing System Simulation Experiments (OSSE) (Cordoba et al., 2017)"* |
| **Line 52**: Use the abbreviation of Observing System Simulation Experiment. | "OSSE framework" |
| **Line 58**: Use the abbreviation of Atmospheric Motion Vectors. | *"uncertainty of AMVs derived from cloud movement ..."* |
| **Line 427**: No need to repeat almost the exact same sentence as in **Lines 43-45.** |  |

---

## Author Comment (AC1)

Manuscript: AMV Error Characterization and Bias
Correction by Leveraging Independent Lidar Data: a
Simulation using OSSE and Optical Flow AMVs

February 2024

**1 Responses to Reviewer 1**

**Overall the proposed scheme is interesting and a useful contribution to the field. Up to and including the bias correction scheme it is well described. After that once we get into (I think) a proposed error estimation scheme I found it difficult to understand. I've picked out a few things that were unclear to me, if these things are explained in the text I'm happy to be corrected.**

We would like to thank the reviewer for the kind comments, and for providing constructive inputs which helped improve our manuscript. Please see our detailed responses below.

1. **Quality of English is good, it would benefit from another look through by the authors to fix a few grammatical errors. Examples (not an exhaustive list):**
   **Line 51 an bias-correction should be a bias correction**
   **Line 88 AMV should be AMVs**
   **Line 171: due -¿ due to**
   **Line 193: track -¿ track of**

   Thank you for the comment. We have fixed the issues above, and we have gone over the paper once more to address the grammatical errors.

2. **In section 2.1 it would be worth explaining how the heights were chosen – traditional AMVs are concentrated around 850 and 200 hPa, so 300 hPa is a little low in the atmosphere and there are normally few quality AMVs at 500 hPa.**

The reviewer is correct that traditional cloud-track AMVs tend to be concentrated at levels with high ( 200 hPa) and low ( 850 hPa) clouds. To explain our reasoning, we added the following text in the last paragraph of Section 2.1 as follows:

> "Traditional AMVs used for tracking clouds are typically focused on high-level clouds (at around 200 hPa) and low-level clouds (at around 850 hPa). Tracking mid-level clouds poses a challenge because they are often obscured by high-level clouds. In this OSSE study, we are considering using AMVs derived from sounder-based water vapor retrievals, which are most reliable in the middle troposphere. Furthermore, lidar winds, primarily derived from the UV (Rayleigh scattering) channel, provide retrievals mainly in the middle to upper troposphere where the scattering signal is adequate for returning Doppler information, and the view is less likely to be obstructed by clouds. For these reasons, we opted to perform our OSSE error characterization experiments at the 850, 500, and 300 hPa pressure levels."

3. **In the same section, line 115-6, why only conventional obs + brightness temperatures to initialise, and not all observation types? And on line 114-5, why the 3rd most powerful member?**

The simulation we chose was used as a nature run in a previously published observing system simulation experiment (Posselt et al., 2022). We chose to use the 3rd strongest ensemble member, as it generated a category 5 hurricane, but retained a realistic development time scale and track (as compared with Hurricane Harvey (2017), upon which it was based). We provide additional detail in Posselt et al. (2022) as to the choice of observation types, but in brief we used conventional observations as the measurements from low Earth orbiting operational satellites were so sparse in time that they had little impact on the initialization of the ensemble.

4. **Line 121: I don't know a great deal about tropical cyclones but a drop of 0.4 hPa seems very small and not a 'plunge'**

We apologize for the error. The 'plunge' is actually 40 hPa, but we made an error in the conversion. We have fixed this in the paper. Thank you for the careful read!

5. **Section 2.2: Although you refer to another paper, you could state here which traditional feature tracking algorithm you're referring to and AMV people will know its main strength**

We apologize for the lack of clarity. Our AMV approach relies on tracking water vapor, though the technique is based on optical flow rather than pattern tracking (Yanovsky et al., 2024). More detail on the algorithm is described in the updated Section 2.2, which is now much expanded compared to the previous draft.

6. **Line 128: what do you mean by dense AMVs? If they are derived for every pixel that already implies they are spatially dense, so are they dense in some other way?**

Upon reading the sentence in question, we agree that the word dense was redundant and somewhat confusing. Therefore we have reworded the sentence as "For every pair of images, the optical flow algorithm generated Atmospheric Motion Vectors for every pixel.".

7. **What part of the NatureRun is used to derive the AMVs – the clouds, as in traditional AMV feature matching? Or the humidity surfaces, or something else?**

We use the humidity surfaces to derive the AMV through an algorithm called optical flow. We have updated Section 2.2 to make this clearer. Thank you!

8. **The main source of error in AMVs is generally thought to be height assignment. If the optical flow is done on WRF humidity fields, then the error is only coming from the tracking and likely to be smaller than typical AMV errors**

We agree with this comment. The height assignment error is a predominant source of error, and in this paper we have tackled the easier topic of fixed-height error. We intend to expand the OSSE study to examine the height-assignment errors in a different paper, and we have noted the difficulties of height-assignment in the following passage in Section 1:

> "Velden and Bedka (2009) along with Salonen et al. (2015) have highlighted the significant impact of height assignment on the uncertainty of Atmospheric Motion Vectors (AMVs) derived from cloud movement and sequences of infrared satellite radiance images. However, this error source is intertwined with uncertainties in the water vapor profile itself, and modeling this within the OSSE framework requires extensive knowledge and parameterization of the height-assignment error process, which is beyond the scope of this paper. As such, in this paper we will focus on fixed-height errors in the AMV estimates and the bias-corrections arising therefrom. "

9. **On figure 1 a scale would be helpful, to know how fast the winds are, and how large the differences are. The factor by which the differences are magnified should also be stated.**

   Thank you for the comment. We have included a legend key (in red) on the top of each plot that should help readers decipher the speed in m/s of each arrow.

   Upon careful consideration, we found that we can make Figure 1 more informative by keeping all the arrows (i.e., optical flow, NatureRun, and difference plots), on the *same* scale. This way, the difference plots will be able to highlight *both* the changes in direction and magnitudes of the windspeed difference.

   The updated Figure 1 is now much improved, in our opinion. Thank you for the feedback.

10. **Section 3.1 I was a bit confused here, are we treating the WRF winds as truth and trying to bias correct towards WRF? Or are we simulating some Aeolus-style u-winds from WRF and bias correcting towards them? How are the lidar winds simulated? There should be some error associated with such lidar winds.**

    We are opting towards the second option: simulating Aeolus-style u-winds from WRF and bias correcting towards them. We are simulating lidar winds by randomly sampling 1% of the data domain as lidar data locations. Another referee also commented on the need to add some error associated with the lidar winds, and thus to address this we opted to add a random zero-mean Gaussian error to the WRF u-wind. The standard deviation for this Gaussian error depends on the pressure levels: 2 m/s for 850 hPa, 3 m/s for 500 hPa, and 5 m/s for 300 hPa.

    After we have added these random errors to the simulated lidar winds, we found that

    - The bias-correction performance (Table 4 and 5) did not change significantly. This is likely because random noise is Gaussian, and their impact is greatly reduced since we are only estimating the first moment (the mean value) in the bias-correction exercise.
    - The coverage percentage in Figure 5 tends to be increased compared to the last draft. This is because we added a constant error (2 m/s for 850 hPa, 3 m/s for 500 hPa, and 5 m/s for 300 hPa), which is added to both the numerator and the denominator of the coverage probability calculation in Figure 5.
    - The increased variability introduced by the lidar simulated error weakens the linear relationship between the predicted error and the empirical error in Figure 6 (i.e., the $R^2$ value is decreased). However, the monotonically increasing relationship is still evident.

- The relationship between the predicted error and Root-Mean-Squared-Vector-Difference (RMSVD) is no longer clear, since the magnitude of the error we introduced (2 m/s for 850 hPa, 3 m/s for 500 hPa, and 5 m/s for 300 hPa for *both* the u and v winds) is too large compared to the magnitude of the typical bias variability ($\tilde{1}$-2 m/s). Therefore, we have opted to remove the subsection on the comparison of the predicted error to RMSVD.

We have updated the corresponding Tables and Figures to reflect the added error for lidar simulated values. Overall, although the strength of some of the relationships have weakened, the conclusions from before (with no error added) are still valid.

11. **What is meant by wind-moisture gradient? Also, I've never heard the word heteroskedasticity and had to look it up, could you define it in the text or explain what you mean without using it? What is 'error residual'?**

We apologize for the lack of clarity. The wind-moisture gradient angle is a short name for the angle between the wind direction and the water vapor gradient. We have opted to remove this short name entirely and just refer to it in full.

Heteroskedasticity, also known as heteroscedasticity, occurs in statistical analysis when the standard deviations of a predicted variable vary across different values of an independent variable, rather than remaining constant. This is an important term because many statistical approaches assumes constant error (homoskedasticity), so recognizing when we have non-constant error variance is important with regards to the selection of the appropriate modelling approach. The 'error residual' here meant 'windspeed difference', and we can see why it was confusing. Therefore, we clarified the sentence in question as follows:

"As for the predictor variables, Posselt et al. (2019) examined the relationship between 'tracked' and 'true' wind using an OSSE framework for the same ETC region as this study, and they noted that there is considerable heteroskedasticity (i.e., non-constant variance) in the windspeed difference (i.e., 'tracked' windspeed minus 'true' windspeed) as a function of water vapor, wind speed, water vapor gradient, and the angle between wind direction and water vapor gradient (Figure 6, Posselt et al., 2019)."

12. **Figure 2 caption – mention that the labels are defined in Table 2**

Thank you for the suggestion. We have added the following lines to the end of the caption of Figure 2: "Variable names along the x-axis are defined in Table 2."

13. **Table 3 caption, mention that the numbers in arrays are being searched through.**

    We have modified the caption of Table 3 to include the following lines: "Note that the arrays under Parameter Settings specify the option grid through which GridSearchCV is searching for the optimal choice." Thank you for the suggestion!

14. **Section 3.2: Line 238-245: I appreciate that there is an attempt to explain MLP but there are lots of new terms that are not fully explained for example nodes and neurons, backpropagation, gradient descent**

    We apologize for the confusion. Unfortunately the discipline of machine learning (specifically that focused on neural networks) have, over their long development, come up with technical terms that are very difficult to explain without large expository materials.

    We have, however, read the excellent tutorial series by Randy Chase– "A Machine Learning Tutorial for Operational Meteorology" (Chase et al., 2022, 2023), and we think this would be a good primer for readers who are not familiar with the machine learning approaches in Section 3.2. Therefore, we have included the following recommendation at the beginning of Section 3.2:

    > "...For this reason, we have chosen four methods that are known to do well for high-dimensional problems with complex relationships: random forest, gradient boosting trees, multi-level perceptron, and nearest neighbor. Here, we touch briefly on an overview of the methods, before going into details of optimization and comparison. For readers who are not familiar with these machine learning approaches, we recommend the excellent tutorial series "A Machine Learning Tutorial for Operational Meteorology" (Chase et al., 2022, 2023)."

15. **Line 250: what is the standard Euclidean distance?**

    The standard Eucledian distance between two points in Euclidean space is the length of the line segment between them. It's also called the Pythagorean distance.

16. **Figure 3 – what is NR?**

    NR is an abbreviation for NatureRun. To make this clearer, we included the abbreviation when we first introduced NatureRun as follows:

"The evaluation of the impact of bias-correction on optical flow AMV will be carried out in the context of an OSSE. All OSSEs share these key components: 1) A reference dataset, used as a basis for comparison. In our case, this is a NatureRun (NR), which is a high-fidelity simulation mimicking real-world conditions..."

17. **Table 4 – what is MarginalSTD and why are some numbers red?**

We apologize for the confusion. We have replaced 'MarginalSTD' with the term 'Uncorrected bias'. And we appended the following sentences in the caption for Table 4:

"...Units are in m/s. A cell that is colored red indicates the best performing method, which is defined as having bias that is closest to zero. The uncorrected bias is defined as the bias of the raw optical flow data relative to the WRF data."

18. **Lines 264-281: by this point we're talking about the bias of the optical flow winds (with or without bias correction applied) vs the WRF simulation truth? Is that right? Perhaps worth re-mentioning the first time 'bias' is mentioned.**

This is indeed the case. We have added the following clarification to the first time bias is mentioned:

"...In all pressure levels, the machine learning approach consistently exhibits smaller bias than the uncorrected optical flow data, where bias is defined as the expected value of the difference between u-wind from optical flow (both corrected and uncorrected) and the WRF simulated truth. "

Thank you for the suggestion!

19. **Line 274: what is meant by state-of-the-art ensemble methods and neural networks? Do any of the methods being tested fit this description?**

We apologize for the confusion. The term 'state-of-the-art ensemble methods and neural networks' refers to the class of algorithm that random forest, gradient boosted trees, and MLP belong to. To remove confusion, we have removed that term entirely and rephrased that sentence as follows:

"...Nearest neighbor, on the other hand, consistently reduces bias relative to the uncorrected optical flow but falls short of the performance achieved by the other three algorithms."

20. **Table 5 – perhaps I missed this in the text, but why are they biases so much larger with the spatial validation than with the temporal validation**

We did not explain this adaquately in the previous draft. The main region is that the functional relationship of the bias and the predictor variables in Section 3.1 probably change more quickly in space rather than in time, hence the difference in performance. We have added the following paragraph around line 310.

> "Another observation is that the validation spatial biases in Table 5 tend to be bigger than the validation temporal biases in Table 4 (e.g., the typical MAB in the spatial case is around .5 m/s, while the typical bias in the temporal validation case is around .05 m/s). In both cases, the machine learning corrected values tend to be improved over the uncorrected optical flow data, indicating that the algorithm is able to capitalize on information within the training dataset for both the spatial and temporal case. However, their difference in performance in Table 4 and Table 5 indicate that functional relationship between the biases and the predictive variables in Section 3.1 may change depending on the spatial region, which makes sense intuitively since different regions of a storm system might exhibit different bias characteristics. However, this functional relationship, as demonstrated by Table 4, tends to be much more stable in terms of temporal evolution in the time scales that we examined (i.e., 3, 4, and 6 hours in advance), which allows the algorithms considered to be more accurate in predicting and correcting the biases. "

21. **Section 3.3 Table 6 – looks like a caption for a non existent table**

That indeed was a reference to another Table that was removed from the last draft. We have removed references to Table 6. Thank you for the careful read. (Please note that deleting the reference to Table 6 now renames Table 7, which are mentioned in Comment 22 and 25, to Table 6 instead.)

22. **Table 7: does it say in the text somewhere which data is Marginal?**

We apologize for the oversight. We have changed the word 'Marginal STD' to 'Baseline STD', and we include the definition of it in the caption as follows:

> 'Figure 6: STD vs Quality Indicators based on RF prediction errors. Baseline STD is defined as the STD of the *entire* optical flow dataset against the WRF simulated truth. (i.e., $\text{std}(\hat{u} - u)$). "

23. **Normally, AMV quality indicators are given as an integer from 0 to 100. Why is a different scale used for these quality indicators?**

   In this section, we aimed to provide a quick demonstration that the predicted uncertainties could be useful for developing quality indicators.

   While normal AMV indicators are typically given as an integer from 0 to 100, our goal is to simplify the analysis to get across the point that the 'High', 'Mid', and 'Low' quality indicators that we developed have the corresponding 'Low', 'Mid', and 'High' errors relative to the withheld truth, as one would expect from the name of those quality indicator categories.

   A mapping of the errors provided by the Quantile Random Forest and the 0-100 AMV Quality Indicators and their performance is of interest, but we feel that this is an important topic that is best examined in detail in a different paper.

24. **Page 19: I found this page quite a struggle, I get that you are trying to come up with error estimates, but what are prediction intervals, nodes, 'leaf' etc?**

   We apologize for the confusion. Unfortunately, the terms 'leafs', 'nodes' refer to the design of random forest, which would require a lot of expository materials that is beyond the scope of this paper. Fortunately, Randy Chase did provide an excellent overview of random forest along with the definition of common terms in the first paper of his series "A Machine Learning Tutorial for Operational Meteorology". Therefore, we have opted to refer the readers to this series again as follows

   > "...In contrast to normal random forests, which approximate the conditional mean of a response variable, quantile random forests (QRF) provide the full conditional distribution of the response variable to construct prediction intervals. (For readers who are not familiar with the random forest algorithm, Chase et al. (2022) provides an excellent meteorology-geared tutorial)"

25. **Is QRF quantile random forest or quantile regression forest?**

   We apologize for the confusion. QRF should mean Quantile Random Forest, and we have gone over the paper and removed all references to quantile regression.

26. **Table 7 is quite a few pages away from where it is referred to, it would be good to move it closer.**

This table has been moved closer. Now it should be presented in the page immediately following when it is mentioned. Thank you.

27. **Section 3.3.1 Line 398: Here it is suggested that the error estimation would only work when lidar data is coincident. I though the idea was to train a model using lidar data, that can then be used any time, anywhere? If the proposed scheme would only work when, for example, Aeolus data happens to be in the same place as AMVs then the scheme would only apply to a very small fraction of AMVs were it used operationally.**

We also wish that this model, after being trained, can be used anytime, anywhere, but unfortunately that is very difficult problem to solve.

In general, machine learning predictions are valid (or useful) when applied to new data that are 'similar' to data that are used in training. Applying a trained model to data that is 'vastly' different from training data is called 'extrapolation', and this is a highly risky procedure that might produce meaningless results.

Part of this paper's focus is to examine what kind of information (and how much) we can extract from coincident AMVs/lidar data. We also examined the limit of 'how far' away from the lidar we can use the trained model. We found that in general, the model is more robust in the temporal direction rather than the spatial direction, and that we can predict as far as 6 hours in advance without significant loss of performance.

Further limits on how 'far' one can go from the lidar data in applying this model is a topic of future research.

28. **Section 4 (Conclusions) Line 450: true AMVs = WRF Nature-Run, correct? Better to say so**

We have changed 'true AMVs' to 'NatureRun wind variability' in this line. Thank you. The line now reads:

> "We observed that, while the prediction intervals often tend to be too narrow (underestimating the variability of the true process), they generally exhibit a monotonically increasing relationship with the NatureRun wind variability."

Again, we would like to thank the reviewer for the careful read and the insightful comments (especially the one about adding random measurement errors to the simulated lidar data, along with the details that we missed earlier such as missing tables and ambiguous captions). The paper has improved significantly after incorporating your feedback!

**References**

Chase, R. J., Harrison, D. R., Burke, A., Lackmann, G. M., and McGovern, A. (2022). A machine learning tutorial for operational meteorology. part i: Traditional machine learning. *Weather and Forecasting*, 37(8):1509–1529.

Chase, R. J., Harrison, D. R., Lackmann, G. M., and McGovern, A. (2023). A machine learning tutorial for operational meteorology, part ii: Neural networks and deep learning. *Weather and Forecasting*.

Posselt, D. J., Wu, L., Mueller, K., Huang, L., Irion, F. W., Brown, S., Su, H., Santek, D., and Velden, C. S. (2019). Quantitative assessment of state-dependent atmospheric motion vector uncertainties. *Journal of Applied Meteorology and Climatology*, 58(11):2479–2495.

Posselt, D. J., Wu, L., Schreier, M., Roman, J., Minamide, M., and Lambrigtsen, B. (2022). Assessing the forecast impact of a geostationary microwave sounder using regional and global osses. *Monthly Weather Review*, 150(3):625–645.

Salonen, K., Cotton, J., Bormann, N., and Forsythe, M. (2015). Characterizing amv height-assignment error by comparing best-fit pressure statistics from the met office and ecmwf data assimilation systems. *Journal of Applied Meteorology and Climatology*, 54(1):225–242.

Velden, C. S. and Bedka, K. M. (2009). Identifying the uncertainty in determining satellite-derived atmospheric motion vector height attribution. *Journal of Applied Meteorology and Climatology*, 48(3):450–463.

Yanovsky, I., Posselt, D., Wu, L., and Hristova-Veleva, S. (2024). Quantifying uncertainty in atmospheric winds retrieved from optical flow: Dependence on weather regime. Submitted for publication to *Journal of Applied Meteorology and Climatology*.

---

## Author Comment (AC2)

Manuscript: AMV Error Characterization and Bias Correction by Leveraging Independent Lidar Data: a Simulation using OSSE and Optical Flow AMVs

February 2024

**1 Responses to Reviewer 2**

**The manuscript explores the application of machine learning techniques to assess bias and uncertainty in the assimilation of atmospheric motion vectors (AMVs). The authors frame the problem by treating independent LIDAR wind observations as a dependent variable in a supervised learning machine model. The study utilizes an Observing System Simulation Experiment (OSSE) framework, with reference geophysical state data derived from high-resolution Weather Research and Forecasting (WRF) simulations.**

**The literature review is comprehensive, providing a strong foundation for the study. The motivation for the research is clearly articulated. However, it's crucial to note that the paper primarily serves as a proof-of-concept, a fact that becomes evident through the text. While the title implies a broader scope, the content remains focused on the proposed machine learning approach for bias correction in wind field assimilation.**

**The approach presented is sound, addressing and resolving issues identified in previous methodologies. The paper is well-structured, and the visual aids effectively support the discussion. However, there are opportunities to better depict certain concepts, as outlined below.**

We would like to thank the reviewer for the kind comments, and for providing constructive input which helped improve our manuscript. Please see our detailed responses below.

1. **The authors should provide a more detailed explanation of their efforts to obtain accurate AMVs. Although they refer readers to another publication for details, as that reference is still "submitted for publication," a general explanation or summary is important for proper understanding.**

We apologize for the lack of clarity. Our AMV approach relies on tracking water vapor using a method called optical flow. More detail of the algorithm is described in the updated Section 2.2, which is now much expanded compared to the previous draft.

2. **Clarity regarding the connection between the proof-of-concept and the utilization of Lidar data is essential. It seems that certain errors associated with Lidar wind profiles were not considered, impacting the comprehensiveness of the study. Clarifying this aspect would strengthen the paper.**

To address this concern, we opted to add a random zero-mean Gaussian error to the WRF u-wind when simulating lidar data. The standard deviation for this Gaussian error depends on the pressure levels: 2 m/s for 850 hPa, 3 m/s for 500 hPa, and 5 m/s for 300 hPa. These are rather conservative numbers since in practice quality filtering can typically reduce the magnitudes of the errors below what are assumed here. However, these somewhat large measurement errors do not adversely affect the conclusions that we have seen in the previous draft.

After we have added these random errors to the simulated lidar winds, we found that

- The bias-correction performance (Table 4 and 5) did not change significantly. This is likely because random noise is Gaussian, and their impact is greatly reduced since we are only estimating the first moment (the mean value) in the bias-correction exercise.
- The coverage percentage in Figure 5 tends to be increased compared to the last draft. This is because we added a constant error (2 m/s for 850 hPa, 3 m/s for 500 hPa, and 5 m/s for 300 hPa), which is added to both the numerator and the denominator of the coverage probability calculation in Figure 5.
- The increased variability introduced by the lidar simulated error weakens the linear relationship between the predicted error and the empirical error in Figure 6 (i.e., the $R^2$ value is decreased). However, the monotonically increasing relationship is still evident.
- The relationship between the predicted error and Root-Mean-Squared-Vector-Difference (RMSVD) is no longer clear, since the magnitude of the error we introduced (2 m/s for 850 hPa, 3 m/s for 500 hPa, and 5 m/s for 300 hPa for *both* the u and v winds) is too large compared to the magnitude of the typical bias variability (˜1-2 m/s). Therefore, we have opted to remove the subsection on the comparison of the predicted error to RMSVD.

We have updated the corresponding Tables and Figures to reflect the added error for lidar simulated values. Overall, although the strength

of some of the relationships have weakened, the conclusions from before (with no error added) are still valid.

3. **The presentation of optical flow could be improved for better interpretation.**

Thank you for the comment. We are addressing this by expanding the section on optical flow in Section 2.2. Please see the answer in #1 for detail of the changes.

4. **The right column in Figure 1, in particular, may benefit from replacing arrows indicating differences with a color-coded scale. Additionally, consider addressing potential confusion related to the arrows' direction by emphasizing differences in magnitude rather than implying directional changes.**

Upon careful consideration of the comment, we found that we can make Figure 1 more informative by keeping all the arrows (i.e., optical flow, NatureRun, and difference plots), on the *same* scale. This way, the difference plots will be able to highlight *both* the changes in direction and magnitudes of the windspeed difference.

We have also included a legend key (in red) on the top of each plot that should help readers decipher the speed in m/s of each arrow.

Thank you for the comment.

5. **While the paper is technically sound, providing a more explicit link between the proposed methodology and Lidar data considerations would enhance the manuscript's overall coherence and contribute to a more comprehensive understanding for readers.**

Thank for the comment. Our goal here is to present a preliminary look at how AMVs could be improved if we knew the wind values for pixels in a lidar orbit curtain that crosses the AMV retrieval area. In particular, we wanted to assess what sort of information we would be able to get from the combination of colocated AMVs and lidar winds.

To this end, we made some simplification in the way we simulated the lidar winds. For instance, we assumed that the simulated lidar wind can only observe the u-wind component, and that we can observe the u-wind with perfect accuracy.

To make it clearer that we are using simulated lidar data in this analysis, we have changed some of the wording in the paper to make it clearer. For instance, in the Introduction, we have modified the description of the experiment as follows:

"We use as our reference (truth, or NatureRun) datasets output from the Weather Research and Forecasting (WRF) Model run for three different weather events (Posselt et al., 2019). The water vapor fields from these WRF model runs are processed through an Optical Flow algorithm (Yanovsky et al., 2024) to provide AMVs, and we similarly simulate lidar observations from the same WRF model data. Finally, we assess the ability of a bias-correction algorithm to model and correct biases (relative to the simulated lidar winds) that arise from the optical flow AMV retrieval."

We also addressed the assumption of using the u-wind component in the simulated lidar wind in Section 3.1:

"It's important to highlight that our results primarily focus on errors related to a single wind component, as lidar systems typically only observe winds along the line-of-sight. In this OSSE study, we simplified the line-of-sight direction to align with the u-wind direction, and we have shown that the uncertainty in the u-wind bias has a positive linear correlation with the validation error. We anticipate that these findings will generalize to the relationship between the HLOS wind and the full-vector wind in other regions, as this is essentially a change of basis for the $(u, v)$ wind components."

The lack of measurement error on the simulated lidar wind (as was in the previous draft) makes it difficult to judge how much the conclusions therein would apply to real world operations. Therefore, we have modified this assumption and added simulated measurement error to the lidar wind as suggested in Comment #2. We believe this modification has strengthened the paper. Thank you for the valuable suggestion!

6. **The first abbreviation of Observing System Simulation Experiments should appear in Line 49.**

   We added the abbreviation (OSSE) on this line. Thank you!

7. **Line 52: Use the abbreviation of Observing System Simulation Experiment.**

   We used the abbreviation here instead of the full name as suggested. Thank you.

8. **Line 58: Use the abbreviation of Atmospheric Motion Vectors.**

   It is fixed as suggested. Thank you.

9. **Line 427: No need to repeat almost the exact same sentence as in Lines 43-45.**

   This sentence ("...NASA's MERRA and MERRA-2 AMVs tend to overestimate wind output by 50% in northwest Europe...") is now removed. Thanks.

Again, we would like to thank the reviewer for insightful comments about adding measurement errors to lidar data, changing Figure 1, and elaborating on the description of Optical Flow. The paper has improved significantly after incorporating your feedback!

**References**

Posselt, D. J., Wu, L., Mueller, K., Huang, L., Irion, F. W., Brown, S., Su, H., Santek, D., and Velden, C. S. (2019). Quantitative assessment of state-dependent atmospheric motion vector uncertainties. *Journal of Applied Meteorology and Climatology*, 58(11):2479–2495.

Yanovsky, I., Posselt, D., Wu, L., and Hristova-Veleva, S. (2024). Quantifying uncertainty in atmospheric winds retrieved from optical flow: Dependence on weather regime. Submitted for publication to *Journal of Applied Meteorology and Climatology*.